# Study on the Expansion and Compression Resistance of 3D-Textile-Reinforced Self-Stressing Concrete

**DOI:** 10.3390/polym14204336

**Published:** 2022-10-14

**Authors:** Xinyu Lu, Boxin Wang, Jiahuan Guo, Tianqi Zhang

**Affiliations:** College of Construction Engineering, Jilin University, Changchun 130021, China

**Keywords:** 3D textile, self-stressing concrete, compressive strength, compression failure mode, strength prediction

## Abstract

Textile-reinforced concrete (TRC), as a kind of high-crack-resistance and high-corrosion-resistance material, has been widely studied. The current research has begun the exploration of the change of textile form, such as 3D-textile-reinforced concrete (3D TRC), and its superior bending performance has been verified. In order to pursue better mechanical properties, combined with the characteristics of self-stressing concrete and 3D textiles, three-dimensional-textile-reinforced self-stressing concrete (3D-TRSSC) specimens were designed in this research. The expansive and compressive properties of specimens with two types of textiles were tested by self-stress and compressibility tests, and the results showed the compressive property and failure mode of 3D-TRSSC were improved compared with 2D-TRSSC and SSC: the increase in compressive strength was 16.3% and 35.1%, respectively. In order to explain the improvement of the compressive strength of the 3D-TRSSC specimens, the triaxial self-stress state analysis of the compressive specimen was carried out, and then a set of calculation methods based on deformation analysis was designed to explain the upward displacement of the necking position of the TRSSC compressive specimen. The theoretical results and experimental data were 27.2 mm and 28–30 mm, respectively. In addition, the improvement of the compressive strength of the 3D-TRSSC specimens relative to that of the 2D-TRSSC specimen was predicted. The calculation results were highly consistent with the predicted values.

## 1. Introduction

In order to seek higher-performance building materials and improve the problems of self-weight, corrosion and poor crack resistance of traditional reinforced concrete materials, textile-reinforced concrete (TRC) has received extensive attention as a high-performance material [1]. Based on previous research, excellent mechanical properties and good environmental adaptability of fiber textiles have been confirmed, and the high crack resistance and high corrosion resistance of TRC have been widely recognized [2]. Different types of fibers have been discussed in detail, and several mainstream kinds of fibers are widely accepted by comparing their performance and price, such as carbon fiber, glass fiber and basalt fiber [3,4]. The current exploration of fiber materials has also confirmed that fibers have a better performance than steel bars. Some composites, such as graphene, can effectively control the content of metal ions in the matrix and improve friction performance [5,6]. Advanced materials are applied in many fields [7,8], and the introduction of advanced materials to concrete is an inevitable trend. For obtaining higher crack resistance, the concrete matrix is also studied. Self-stressing cement can produce expansion deformation in the hydration process, if constrained, and a certain degree of self-stress can be generated in the matrix to improve crack resistance. In addition, because of the characteristics of a small diameter and the light weight of textiles, it has a certain guiding significance for making lightweight concrete specimens [9]. The ability of textiles to adapt to the environment is much higher than that of steel bars, which makes TRC work better than traditional reinforced concrete in complex environments, such as freeze–thaw cycles, salt corrosion and other environments [10].

Research on textile forms mainly focuses on two-dimensional textiles, and numerous kinds of specimens and failure modes of 2D-textile-reinforced concrete (2D TRC) have been studied, such as debonding failure, bending failure and shear failure. A lot of conclusions have been obtained regarding the distribution rate, laying position and laying method of 2D textiles [11,12,13]. The fiber layers and fiber arrangement in 2D TRC have been tested, and numerous studies have been carried out on the bending properties, crack resistance and tensile properties of 2D TRC [14,15,16,17,18,19]. This form of TRC shows excellent working performance and convenience in the process of component reinforcement. However, in the production of stressed components, single-layer textiles do not have spatial stability, layered pouring is needed in the specimen, and attention should be paid to the control of textile deformation in the vibration process. It is not convenient and difficult to ensure the construction quality in the construction process. The latest research has begun to seek higher-performance textile forms. Collaboration fibers are used to connect the single-layer textiles, making the two-layer textiles become a fiber cage, and the 3D textiles with space stability has been made. The specimen that is made of this form of textiles is called three-dimensional-textile-reinforced concrete (3D TRC) [19]. In addition to focusing on the excellent bending properties of 3D TRC, Eliyahu Adiel Sasi compared the effects of different types of fibers and epoxy resin impregnation on 3D textiles [20]. R. Haik also discussed the effect of epoxy resin and tested the effect of non-orthogonal fiber on mechanical properties [21]. Michael El Kadi tested the laying direction and combination of 3D textiles and, in addition, optimized the production process of three-dimensional textiles [22]. According to the findings of these scholars, this form of textile has excellent self-stability, which can ensure the stability of its position in the process of production and pouring without the tedious process of layered arrangement. It can not only simplify the production process of the specimen, but also improve the pouring quality of the specimen and further improve its working performance.

Referring to the influence of self-stress on the concrete matrix, 3D TRC and self-stressing concrete (SSC) were combined to make three-dimensional-textile-reinforced self-stressing concrete (3D-TRSSC) specimens in this paper. The self-stress value test and the compression test were designed to explore the compressive performance and failure mode of 3D-TRSSC, 2D-TRSSC, SSC and ordinary Portland cement concrete (NC) [23]. The test results showed that the compressive performance of SSC was worse than that of NC, and the two forms of textiles had little effect on the compressive strength of NC, but the addition of textiles could effectively improve the compressive strength of SSC, and 3D textiles could increase the compressive strength of SSC to the level of NC. Compared with 3D-TRSSC and 2D-TRSSC, it was found that the addition of collaborative fibers not only effectively improved the compressive strength of the specimens, but it also improved the compressive failure mode. In this paper, through the analysis of the deformation of the specimen, a set of calculation methods for the increase in compressive strength of 3D-TRSSC compared with 2D-TRSSC was obtained, and the calculation results were in good agreement with the experimental results.

## 2. Materials and Methods

### 2.1. Matrix

Two kinds of cement, sulphate aluminum cement (level 4.0 stress, self-stress strength grade of 28 days, Wuxi Jianghuai Building Materials Technology Co., Ltd, Wuxi, Jiangsu, China) and ordinary Portland cement (P. O 42.5, Wuxi Jianghuai Building Materials Technology Co., Ltd, Wuxi, Jiangsu, China), were used in the experiment. The diameter of limestone coarse aggregate was 8 to 16 mm, and the fineness (an index of particle size) modulus of fine aggregate river sand was 2.5 (Changchun Building Materials Wholesale, Changchun, Jilin, China). Sika-3301 mh. Third-generation polycarboxylate superplasticizer (SIKA (China) Co., Ltd, Suzhou, Jiangsu, China) was added to the specimens to ensure the fluidity of the concrete matrix, and the cement dosage was more than 550 kg/m^3^ to ensure the self-stress value [4]. All specimens were cured for 28 days after hydration for 24 h. The mixture ratio is shown in Table 1.

### 2.2. Textile

All textiles were made of carbon fiber and by the method of plain weaving. Single-strand fiber twistless roving was twisted to the form of fiber bundles, then warped and weft fiber bundles were interspersed up and down to obtain the flat single-layer textile. The fiber model was T700SC-24000, produced by Toray Corporation of Japan(Toray Industries(China)Co.,Ltd. (TCH), Nantong, Jiangsu, China). There was only a difference in Tex (a linear density unit) content between the two selected carbon fibers. The specific mechanical parameters are shown in Table 2.

The experiment mainly involved two kinds of textile types and two kinds of grid size: a 2D textile (with a grid size of 20 × 20 mm and 30 × 30 mm) and a 3D textile (with a grid size of 20 × 20 × 20 mm and 30 × 30 × 30 mm). All specimens had a concrete cover thickness of 20 mm.

The fabrication of the 3D textiles was based on the 2D textiles, which were produced by traditional plain weaving. The corresponding nodes of the two-layer textiles were connected by two collaborative fibers, and the connection method was the diagonal connection to a form an X-like shape, and this approach effectively and efficiently connected the two corresponding nodes to obtain a stable spatial structure. In order to ensure the stability of 3D textiles and the equal Tex content of each side, the Tex content of the collaborative fiber was 800, and the Tex content of the planar textile was 1600. All textiles were impregnated with Phoenix WSR6101 (E-44) epoxy resin (QIANGUANG CHEMICAL INDUSTRY, Wuxi, Jiangsu, China) to ensure the stability of the textile [23]. The production method of the textiles is shown in Figure 1.

### 2.3. Self-Stress Value Test

In order to ensure the measured self-stress value is more representative, it is necessary to limit the whole section of the specimen to measure a reliable full section of self-stress [24]. The self-stress value test specimen was designed as shown in Figure 2. The size of the specimen was 100 × 100 × 400 mm. The two ends of the steel bar were connected to the two baffles by a welding method as a whole, and they were poured into the self-stressing concrete test specimen. The deformation of the steel bar represented the average deformation of the concrete matrix. The steel bar with an elastic modulus of 2.0 × 10^5^ and a diameter of 6 mm was selected [23].

The specimen was maintained in the maintenance barrel for 28 days, and the test lasted for 28 days, too. During the measurement, the displacement of the baffles was recorded in real time by a displacement meter until the maintenance was completed, and the maximum deformation was recorded.

### 2.4. Compression Test

Compared to Portland cement, self-stressing cement produces ettringite (AFT) crystals during hydration, resulting in a decrease in matrix density and strength. However, at the same time, if the expansion is limited, a certain degree of self-compression stress can be generated within the matrix: for example, self-stress is produced under the constraint of the textile in TRSSC. Under the influence of 3D textiles, the internal concrete is under three-way pressure and the self-stress direction is distributed along the fiber direction. In general, this stress state would have a large impact on the compressive strength. Thus, the influence of different cement matrices on the compressive failure mode and compressive strength had a certain reference significance for the design of specimen grouping in the compressive test. A textile was set in each group of specimens to control the textile effect on the strength of concrete during the compression test. The elements of the compression test in the control group included the cement type, textile grid size and textile type. The ordinary concrete specimens and self-stressing concrete specimens with the same mixing ratio were tested and, on this basis, 2D and 3D textiles with a grid size of 20 mm and 30 mm were added, respectively. The size of each specimen was 100 × 100 × 100 mm.

The textiles in both kinds of specimens were arranged with the warped and weft fibers parallel in the horizontal direction. For 3D textile specimens, the distance between the textile and the bottom of the specimen was 20 mm, and for the 2D textile specimens, the distance between the bottom layer of the textile and the bottom of the specimen was also 20 mm to ensure that the position of the textile in each sample remained unchanged. Due to the particularity of the 2D-TRSSC specimen, it is necessary to make a layered pouring mold to fix the position of textiles. The mold was designed to be able to be disassembled in the position of the textile, and the textiles were clamped and fixed. Because of the confinement of the fiber bundles, compressive stress was generated inside the concrete matrix in the direction of the fibers. There were six specimens in each group. The whole process was conducted on a servo-controlled hydraulic actuator (DNS300, Changchun Research Institute for Mechanical Science Co., Ltd., Changchun, Jilin, China) with a displacement control of 0.2 mm/min. The specific compression test group is shown in Table 3.

## 3. Results and Discussion

### 3.1. Expansion Properties

The expansion of self-stressing concrete is reflected in its self-stress value. There is a force equilibrium relationship in the specimen as shown in Equation (1), and the self-stress value of the concrete matrix could be calculated by the strain of the steel bar. Comparing the self-stress value measured by textile constraints, it could be found that the results obtained by the two methods were similar under the same mixture ratio. Test data are shown in Table 4.



(1)
$$A_{s}\varepsilon_{s}E_{s}=A_{c}\sigma_{\mathrm{ss}}$$

*A*_s_: cross-sectional area of steel bar*ε*_s_: strain of steel bar*E*_s_: modulus of elasticity of steel bar*A*_c_: cross-sectional area of concrete*σ*_ss_: self-stress value of self-stressing concrete


### 3.2. Compression Properties

The specific compression test data are shown in Table 5. By analyzing the average value of compressive strength, group 8 was 22.5% lower than that of group 7, while group 4 was higher than group 2. This means that the use of 3D textiles could effectively improve the compressive strength of self-stressing concrete. Compared to the control group with different textile arrangements under the self-stress matrix, when the grid size of the textile was 20 mm, results showed that the compressive strength of 3D-TRSSC increased by 16% compared to 2D-TRSSC, and 35% to SSC. However, when the grid size came to 30 mm, the increase between 3D-TRSSC and 2D-TRSSC was only 8%. Furthermore, in the same type of textile, the strength of the 20 mm textile was also about 5–13% higher than that of the 30 mm textile, and the gap between the 3D textiles was bigger. After removing the highest and lowest data in each group of experiments, the load–displacement curve is shown in Figure 3.

The data revealed that the type of textile had little effect on the strength of Portland concrete because the expansion did not occur, and the textile cannot direct self-stress in the matrix. However, in the self-stressing concrete matrix, considering that the self-stressing matrix could produce compressive self-stress under the constraint of the textile, the concrete in the core area was naturally in a two- or three-way compression state within the influence of self-stress. As a result, the compressive strength of self-stressing concrete under multi-axial compression could be improved.

To explain the enhancement of compressive strength by 3D collaborative textiles, the core area of the specimen was assumed to be in a triaxial compression state, and the axial pressure was provided by the compressive self-stress. Triaxial compression analysis was carried out as follows: 0 > *σ*_1_ = *σ*_2_ > *σ*_3_, *σ*_1_ = *σ*_2_ = 2 MPa. The stress in the directions of *σ*_1_ and *σ*_2_ was the self-compression stress inside the matrix, taken as 2 MPa. Axial compression was much lower than compressive strength, and it belonged to the conventional triaxial compression state. The uncorrected Richart formula [25,26] is applied for calculation:(2)$$\frac{f_{3}}{f_{c}}=1+\frac{4.1f_{1}}{f_{c}}$$*f*_3_: conventional triaxial compressive strength*f*_c_: compressive strength of self-stressing concrete considering textile restraint but not self-stressing state*f*_1_: the self-stress value in the direction of *σ*_1_ and *σ*_2_


Thus, we achieved *f*_3_ = 1.24 *f*_c_ by the above calculation. Compared to the results of the test and referring to the influence of textiles on compressive strength in NC, *f*_c_= 31.53 MPa. Group 4 (20 mm 3D-TRSSC) yielded *f*_3_ = 1.35 *f*_c_, while Group 6 (30 mm 3D-TRSSC) yielded *f*_3_ = 1.19 *f*_c_, compared to the 3D self-stress state with 3D collaborative textiles. Theoretically, the relationship between triaxial compressive strength and biaxial compressive strength was *f*_3_ = 1.15 *f*_2_ in this state, but the relationship between 30 mm 3D-TRSSC and 30 mm 2D-TRSSC was 2 *f*_3_ = 1.11 *f*_2_. Notably, the distribution of compressive self-stress in the matrix was limited; thus, the specimens cannot be considered the perfect triaxial compression condition that is completely covered by the action range of compressive self-stress, and the compressive self-stress decreased with the development of lateral axial expansion. However, the actual value was significantly higher than the calculated value in the case of the 20 mm grid. Given such information, it is also necessary to explore the synergistic effect of the 3D textile in conjunction with the failure mode. The failure process of the 20 mm 3D-TRSSC specimen observed in the test is shown in Figure 4.

As can be seen, each side of the specimen was divided into three small columns whose lengths ranged from 20 to 50 mm during the failure process, and the main crack that separated into small columns was wide. In the test, the failure mode of multi-axial compression, which was different from the form of the positive and negative connected quadrangular pyramid, could be observed. At the same time, the failure toughness of the test block was improved to a certain extent. However, with the lateral expansion and crack of the specimen, the lateral self-stress was consumed, and the specimen showed a trend of shifting from triaxial compression to excessive axial compression. As shown in Figure 5, the specimen continued to load after reaching the ultimate compressive strength. Then, with the widening of the longitudinal main crack, the axial self-stress gradually disappeared, and the failure mode of the test piece gradually developed to the mode of the positive and negative connected pyramid. Furthermore, the textile had a limiting effect on the expansion of the concrete matrix, and the final failure mode roughly reflected the form of the positive and reverse connected pyramid. However, the retention rate of the textile part was higher, the two pyramids were not symmetrical, and the necking phenomenon was clearly improved.

The 3D equivalent self-stress in the matrix was introduced by the 3D collaborative textile. The concrete between the textile layers was in the state of 3D compression under the level of self-compression stress. The self-stressing environment of 3D compression could prevent the occurrence of interface cracking, and the failure mode changed from tension cracking or splitting to cement-slurry crushing. Meanwhile, the 2D textile could not give full play to the advantages of self-stress due to the same self-tress value in just two directions and because the improvement of the matrix strength was relatively small. The results showed that the 3D textile could improve the compressive strength of the matrix by 35%, along with the compression failure mode of the matrix and the failure toughness and other indicators. Then, in comparison with the 2D textile, the improvement of compressive strength of the 3D textile specimens exceeded the influence range of triaxial self-stress, so it is assumed that the collaborative fibers within the 3D grid had an impact on the textile after the deformation of the specimen, and the improvement was found to be about 16% by comparing the test data with the calculated results. This indicated that the presence of 3D self-stress and 3D confinement not only affected the stress state of the concrete in the core, but also had an impact on the development after cracking.

In order to explain the excessive increase in the compressive strength of the 3D textile specimens relative to the 2D textile specimens, the role of collaborative fibers needed to be analyzed in the context of the necking phenomenon. In the process of compression, by observing the failure mode, it could be found that, due to the transverse expansion of the specimen, the position below necking was under the tensile stress state. After the formation of the penetration crack, the tensile region was deformed significantly. Assuming that the necking position reached the ultimate tensile strain, the strain relationship is shown in Figure 6.

It is assumed that, after the occurrence of cracks, the difference of deformation between the two layers of textiles is considered, and that the specimen has produced a large deformation, resulting in the collaborative fibers between the layers no longer remaining vertical and becoming deformed. The collaborative fibers in the 3D textile begun to exert an effect on the two layers of textiles, and since the node was fixed by epoxy resin as a whole, the collaborative fiber adjusted the internal force of the upper and lower two layers of the textiles through deformation. The necking position usually occurs at the geometric center of the specimen in plain concrete; however, due to the addition of textiles, the deformation of textiles shared the stress at its location, resulting in a change in the necking position, if equal deformation energy is considered to be generated on both sides of the necking position to maintain the stability of the specimen. In the process of concrete compression, lateral expansion occurs, resulting in tensile stress in the lateral direction of the concrete. The height of the necking position is determined by the deformation energy on both sides of the necking. It is assumed that the energy balance generated by the deformation on both sides of the necking position can obtain the equation about the height of the necking position as shown in Equation (3).
(3)$$\left(\frac{h-h_{2}}{t}+\frac{h-h_{1}}{t}\right)\cdot \varepsilon_{\mathrm{tu}}\cdot E_{s}\cdot D+\int_{h_{1}}^{h_{2}}\frac{\varepsilon_{\mathrm{tu}}}{t}\cdot xdx\cdot E_{c}=\int_{0}^{h_{0}}\frac{\varepsilon_{\mathrm{tu}}}{h_{0}}\cdot xdx\cdot E_{c}$$
*E*_s_: modulus of elasticity of fibers*E*_c_: modulus of elasticity of concreteε_tu_: ultimate tensile strain of concrete*t*: distance from necking position to the bottom of the specimen*D*: diameter of fiber bundle, 1 mm


In the case of this test, the calculated result was obtained as *h*_0_ = 27.2 mm, and by observing the actual text, *h*_0_ = 28–30 mm: the calculated result is in good agreement with the actual situation. The role of collaborative fibers could be analyzed through the deformation relationship after the necking position is determined. A certain angle existed in the collaborative fibers because the two layers of textiles were not equally deformed. In addition, the horizontal deformation of each section of the whole specimen could be obtained by calculating the necking position, and the stresses in the two layers of the textile would be adjusted as shown in Figure 7.

Deformation occurred in the collaborative fibers, along with stress. As a result, the internal forces of the textile were adjusted by the collaborative fibers. Assuming that the specimen maintained the strain relationship, no damage occurred at the nodes of the 3D textile, and there was no slippage of the textile with the concrete. By calculating the effect of collaborative fibers on the textiles, it is possible to analyze the improvement of 3D textiles on 2D textiles. The procedure for calculating the degree of improvement is shown in Equation (4)–(7).
(4)$$\varepsilon_{1}=\frac{\varepsilon_{e}\cdot \left(h-h_{1}\right)}{h-h_{0}};\varepsilon_{2}=\frac{\varepsilon_{e}\cdot \left(h-h_{2}\right)}{h-h_{0}}$$
(5)$$\tan\theta =\frac{b\left(\varepsilon_{1}-\varepsilon_{2}\right)}{2a}$$
(6)$$F=\left(\frac{1}{\cos\theta }-1\right)\cdot E_{s}$$
(7)$$\rho =\frac{2F\cdot \sin\theta }{\varepsilon_{1}E_{s}}=\frac{b\cdot \left(h_{2}-h_{1}\right)}{a\cdot \left(h-h_{1}\right)}\cdot \left(1-\frac{1}{\sqrt{1+\tan^{2}\theta }}\right)$$
*ε*_e_: ultimate expansion strain in the lateral direction*b*: width of the specimen*a*: spacing between textiles*ε*_1_: strain of the upper textile layer*ε*_2_: strain of the lower textile layer*θ*: rotation angle of collaborative fiber*F*: internal force of collaborative fiber*ρ*: percentage of increase


ε_e_ is taken as 0.15 according to the test, and if the experimental data of 20 mm 3D-TRSSC and 20 mm 2D-TRSSC were taken into account to calculate, it could be obtained that *ρ* = 15%. If the measured data of 3D-TRSSC and 2D-TRSSC with a grid size of 30 mm were brought into the above method, the improvement degree *ρ* = 10% could be obtained. This result is in good agreement with the experimental results, which proved the reasonableness of this calculation method and explained the change of compressive strength of the TRSSC specimen by changing the two forms of textiles.

## 4. Conclusions

In this paper, the fabrication of 3D textiles and the basic mechanical properties of 3D-TRSSC were studied, and the reasons for the improvement of mechanical properties of 3D-TRSSC compared with 2D-TRSSC are explained. It provides a new idea for the fabrication of TRSSC and provides a reference for the fabrication of TRSSC stress components and reinforcement components.

The 3D self-stress and 3D restraint effect introduced by the 3D collaborative textile can effectively ameliorate the compressive performance of the specimen and improve the compression failure mold form of the specimen. By observing the experimental data, due to the combined effect of 3D self-stress and collaborative fibers, the compressive performance of the 3D-TRSSC was improved by 16.3% compared to the 2D-TRSSC and 35.1% compared to the self-stressing concrete;The self-stress value was the same under the condition of using baffles and textile. Using this self-stress value and the triaxial stress model, the compressive strength of 3D-TRSSC was 24% higher than that of SSC, and the compressive strength of 2D-TRSSC was 15% higher than that of SSC in theory;A set of calculation methods for calculating the failure mode of 3D-TRSSC under compression and the increase in compressive strength of 3D-TRSSC compared with the 2D-TRSSC specimens was proposed. By calculation, the height of the neutral axis is 27.2mm, and the compressive strength of 3D-TRSSC is increased by 15% compared with that of 2D-TRSSC.

## Figures and Tables

**Figure 1 polymers-14-04336-f001:**
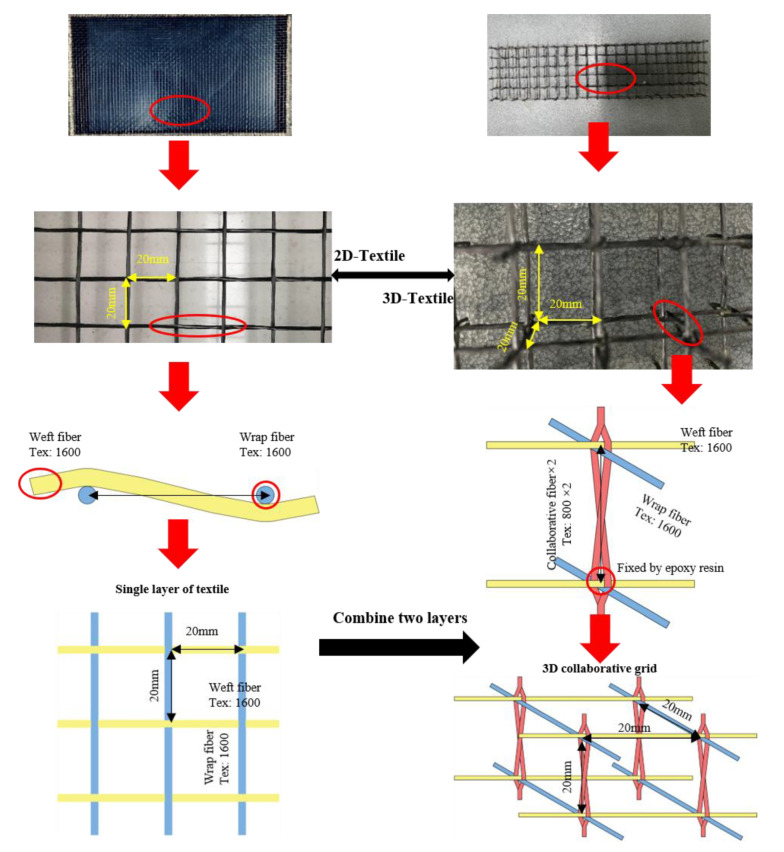
Details of different types of textiles.

**Figure 2 polymers-14-04336-f002:**
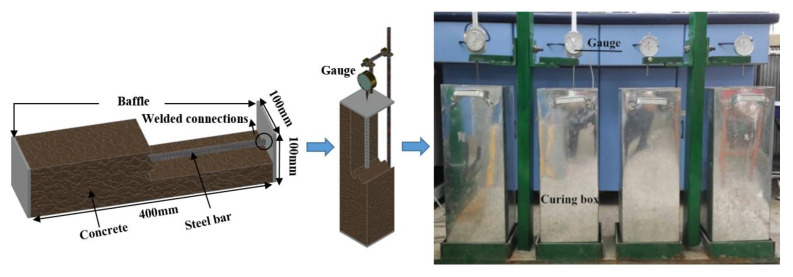
Process of self-stress value test.

**Figure 3 polymers-14-04336-f003:**
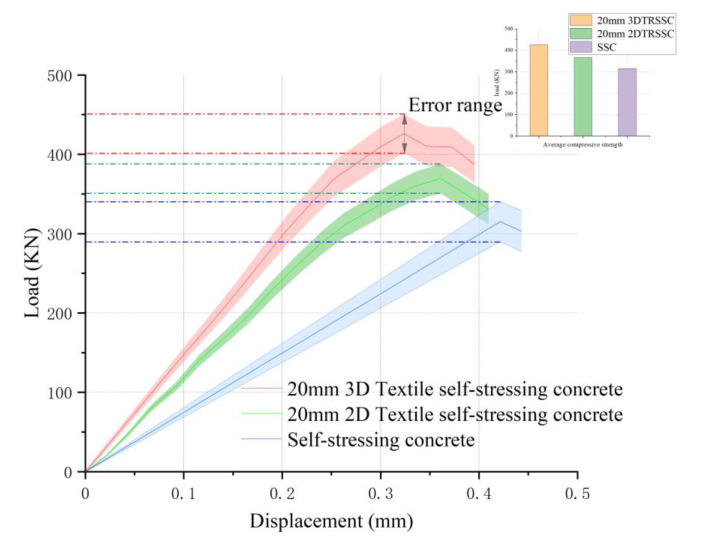
Load–displacement curve of compression test.

**Figure 4 polymers-14-04336-f004:**
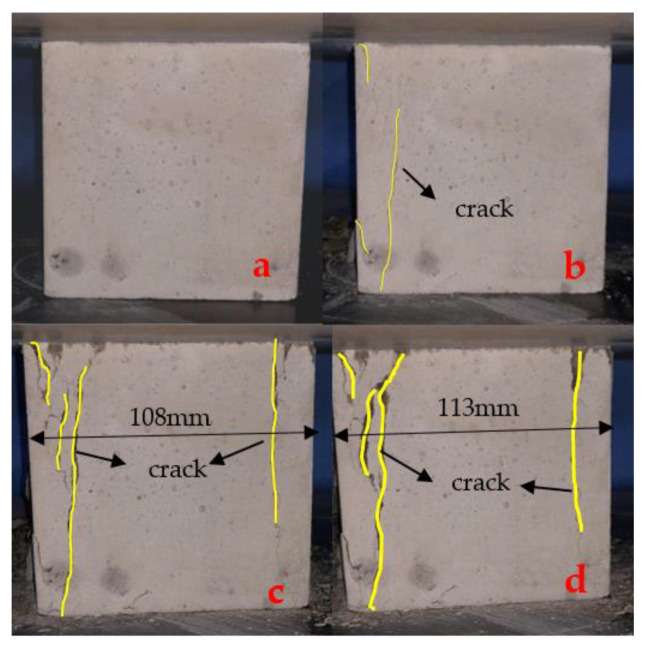
Compression failure process of 3D-TRSSC. (**a**) Pre-cracking; (**b**) initial cracking; (**c**) development of cracking; (**d**) limit of cracking.

**Figure 5 polymers-14-04336-f005:**
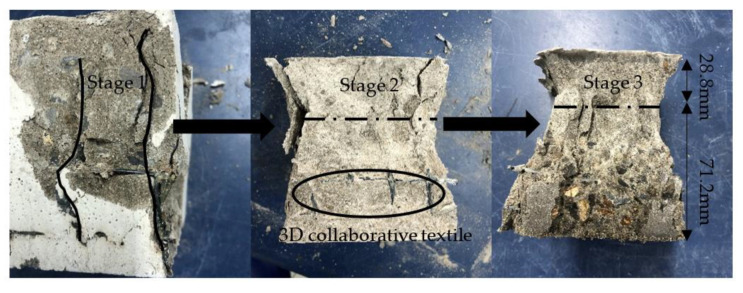
Failure mode of 3D-TRSSC ultimate strength.

**Figure 6 polymers-14-04336-f006:**
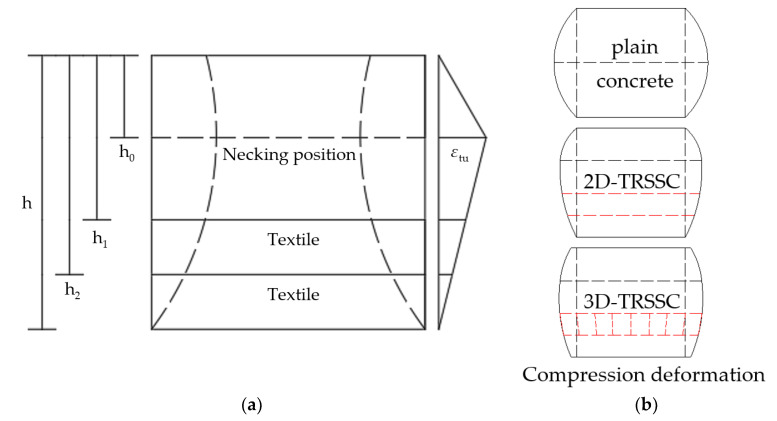
(**a**) Strain relations in compressive specimens; (**b**) the macroscopic deformation diagram of specimen under compression.

**Figure 7 polymers-14-04336-f007:**
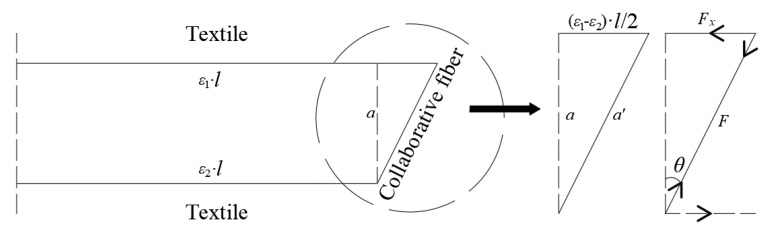
Deformation of 3D textile during compression.

**Table 1 polymers-14-04336-t001:** Mixture ratio of concrete (kg/m^3^).

Matrix	Type of Cement	Cement	Water	Fine Aggregate	Coarse Aggregate	Super Plasticizer
NC	Portland cement	663	239	796	530	2.65
SSC	Sulphate aluminate cement	663	239	796	530	2.65

**Table 2 polymers-14-04336-t002:** Mechanical parameters of carbon fiber bundle.

Tensile Strength (MPa)	Tex (k)	Elastic Modulus (Gpa)	Density (g/cm^3^)	Theoretical Sectional Area (mm^2^)	Poisson’s Ratio
4900	1600	230	1.76	9.09	0.307
4900	800	115	1.76	2.27	0.307

**Table 3 polymers-14-04336-t003:** Specimens for compression experiments.

Group	1	2	3	4	5	6	7	8
Matrix	NC	NC	SSC	SSC	SSC	SSC	NC	SSC
Type of textile	Double- layer	3D	Double- layer	3D	Double- layer	3D	/	/
Size of textile (mm)	20	20	20	20	30	30	/	/

Note: NC-ordinary Portland cement; SSC-sulphate aluminum cement.

**Table 4 polymers-14-04336-t004:** Data of the self-stress value test.

Type of Constraint	Deformation of Reinforcement (mm)	Strain Value of Reinforcement (×10^−3^)	Self-Stress Value (mPa)
Steel bar and baffles	0.79–1.83	1.97–4.58	1.12–2.59
Textile	/	/	1.10–2.50

**Table 5 polymers-14-04336-t005:** Average compressive strength of each group of specimens.

Group	Type of Matrix	Type of Textile	Size of Textile (mm)	Average Compressive Strength (mPa)	Standard Deviation
1	NC	Double- layer	20	40.97	1.13
2	NC	3D	20	40.77	2.25
3	SSC	Double- layer	20	36.63	0.44
4	SSC	3D	20	42.60	1.28
5	SSC	Double- layer	30	34.87	2.88
6	SSC	3D	30	37.67	1.12
7	NC	None	0	38.62	2.30
8	SSC	None	0	31.53	1.89

## Data Availability

All data are included in the paper or in the supplementary data.

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
