# Peer review of "Study on the Expansion and Compression Resistance of 3D-Textile-Reinforced Self-Stressing Concrete"

_polymers, 2022, doi:10.3390/polym14204336_

Round 1
Reviewer 1 Report
[Study on The Expansion and Compression Resistance of 3D 2 Textile Reinforced Self-Stressing Concrete]
Authors studied the deformation behaviors of 3D textile reinforced self-stressing concrete and its failure along with 2D-TRSSC and SSC. There are several points needed to be addressed for consideration of the publication.
1. (Line 105-107) In this work, two kinds of grid size(20*20, 30*30mm for 2D/3D cases) are only considered and tested. And then authors concluded the deformation behaviors of the general 3D/2D SSC based on such two cases. It is well known that the properties of the composite materials could significantly depends on their fiber/matrix configuration and spacing. Test results could be different for other grid size cases which are not considered in this work.
2. Typo (Line 179) “As” probably needs to be modified as “Ac”.
3.
(1) In table 5, three different factors(type of matrix, type of textile, size of textile) were investigated for their effects on compressive strength, but only 8 cases were considered for such investigations. Those are too small number to conclude the correlation between specimen types and fracture strength.
(2) Also, Compressive strength is not significantly different for different specimen preparation conditions (i.e. 31.53 – 42.60 MPa). On the contrary, the error ranges in Fig. 3 are too big. Based on the table 5 & Fig.3, the findings in line 199-204 cannot be supported strong.
4. (Line 270) “the excessive increase in compressive strength of 3D textile spec-270 imens relative to 2D textile specimens”. As stated above, the compressive strength is not significantly different for the cases considered. Also, the error ranges in Fig. 3 are too big compared with the difference of compressive strength for each cases in Table 5.
5. (1) How the Equations (3)- (7) were derived?
(2) In Eq.(3) the unit of the first term is different from the ones of the second and third terms. Check the equation.
6. Overall, no major results are found in this study.
Author Response
Comment 1: (Line 105-107) In this work, two kinds of grid size(20*20, 30*30mm for 2D/3D cases) are only considered and tested. And then authors concluded the deformation behaviors of the general 3D/2D SSC based on such two cases. It is well known that the properties of the composite materials could significantly depends on their fiber/matrix configuration and spacing. Test results could be different for other grid size cases which are not considered in this work.
Thank you for pointing this out, we have made changes on lines 114-115 page 3. In this paper, we mainly explore the influence of different types and mesh sizes of textiles on the compression properties of the matrix, so there is no variable about the thickness of the concrete cover in the test. All the textile bottom layers are kept 20 mm apart from the matrix, so as to control the influence of the variable of the concrete cover.
Comment 2: Typo (Line 179) “As” probably needs to be modified as “Ac”.
Thank you for your careful observation and correction, we have made changes on line 189.
Comment 3 :
(1) In table 5, three different factors(type of matrix, type of textile, size of textile) were investigated for their effects on compressive strength, but only 8 cases were considered for such investigations. Those are too small number to conclude the correlation between specimen types and fracture strength.
(2) Also, Compressive strength is not significantly different for different specimen preparation conditions (i.e. 31.53 – 42.60 MPa). On the contrary, the error ranges in Fig. 3 are too big. Based on the table 5 & Fig.3, the findings in line 199-204 cannot be supported strong.
(1) We thank you for pointing this out. In the compression properties test, Our main purpose is to explore the influence of different textile types on the matrix, and due to the limitations of the plain weaving method and the size of the specimens, too large textile size will make the upper textile position too high, it difficult to participate in the force process, and too small textile size will affect the continuity of the aggregate. So we designed three variables include the type of matrix, the type of textile, and the size of the textile, and each of these three variables includes two control groups, such as the type of matrix including Potelan cement and sulphate aluminium cement, the type of textile including 2D and 3D, and the size of the textile including 20mm and 30mm. We have designed all eight possible experiments, and in future studies we will continue to study the size effect of textiles in detail by changing the specimen size and textile size.
(2) Thank you for your careful observation. Represented by the average value, the compressive strength of different textile forms can be increased by 35%, the maximum increase between different textile types is 16%, and the maximum increase of different sizes of textiles is 13%. All the control groups adopted the same mix ratio to ensure that the mechanical properties of the matrix are the same. It can be considered that all the improvement is due to the changes of textiles. From the percentage of increase, the increase in compressive strength is considerable. In addition, it can be seen from figure 3 that the maximum compressive strength of 2D textile specimens after considering the error is still below the average compressive strength of 3D textile specimens, and this trend will be more obvious as the sample increases.
Comment 4: (Line 270) “the excessive increase in compressive strength of 3D textile spec-270 imens relative to 2D textile specimens”. As stated above, the compressive strength is not significantly different for the cases considered. Also, the error ranges in Fig. 3 are too big compared with the difference of compressive strength for each cases in Table 5.
Thank you for your careful review, we have made changes on figure 3 and lines 203-204. By observing the data in table 5, it can be found that when the textile size is 20 mm, the compressive strength of 3D TRSSC is 16 % higher than that of 2D TRSSC. This improvement occurs when the basic mechanical properties of the matrix remain unchanged, and the error overlap in Figure 3 is the highest data of 2D TRSSC and the lowest data of 3D TRSSC. If the highest and lowest data of the six test blocks in the test are removed, the trend will be clearer.
Comment 5: (1) How the Equations (3)- (7) were derived?
(2) In Eq.(3) the unit of the first term is different from the ones of the second and third terms. Check the equation.
(1) We thank you for your question, Eq 3 is derived from the variable performance equilibrium relationship on both sides of the necking position, and Eq 4-7 are derived from the textiles' geometrical relationship with the specimen's compressive deformation, as shown in Fig. 7.
(2) We thank you for pointing out this mistake, we have updated Eq 3 more accurately. In the previous calculation, the diameter of the fiber is considered to be 1mm, and the fiber diameter is omitted in the formula. The modified formula has corrected this problem. The theoretical diameter is calculated to be 1.08mm by the parameters of the fiber. In order to facilitate the calculation, the fiber diameter is taken as 1mm in the calculation process.
Comment 6: Overall, no major results are found in this study.
We thank you for your suggestion, as a new form of textiles, the effect of 3D textiles on the properties of the matrix needs to be explored. In this paper, the basic mechanical properties of this new composite material were explored, and the influence of 3D textiles on the compressive failure mode and compressive strength was studied. It has a certain effect on the study of 3D TRSSC stress components, reinforcement components and the production of 3D textiles. And in theory and practice, 3D TRSSC relative to the traditional 2D TRSSC has a more than 15 % increase, relative to the SSC specimen is more than 30 % increase.

Reviewer 2 Report
After reading this mansucript, I rally have doubts about its suitability for Polymers. It seems to me more suitable for a materials engineering journal, as carbon fiber (the basis of the "textile" being tested) is not actually a polymeric material, but since the editor considers this a potential contribution, I detail below my comments:
1. The manuscript, particularly at the beginning contains abundant technical abbreviations which are not necessarily known to the average reader of Polymers: Cement level 4.0 stress; fineness modulus, Tex content,...
2. Improver use of the past tense (very frequent in the manuscript):....was shown in Figure (should be ...is shown...)
3. table 3. the caption is very incomplete
4. Section 3.2 title: compress-> compression
5. Figure 3: caption icnomplete. One needs to go to the text to understand what is shown. Type of cement, meaning of sample names
6. Figure 4: 108 m 113 m, should be mm?
7. Eq. 3 needs more details as to its derivation
8. Table 5: compression strength means compression stress to rupture? Please explain
9. Lines 199-204 are confusing: ...little effect....compresion could be improved. Is there improvement or not?
10. Line 208: why 2 MPa?
11. Lines 222-226 also confusing. please rephrase in amore accessible manner.
12. Figure 4: we see 4 pictures, but nothing is explained in the caption. Label a,b,c,d and explain, please
Author Response
Comment 1: The manuscript, particularly at the beginning contains abundant technical abbreviations which are not necessarily known to the average reader of Polymers: Cement level 4.0 stress; fineness modulus, Tex content,...
Thank you for pointing this out. We have made changes on lines 91-94 and line 107. Cement level 4.0 stress means the self-stress value of self-stress sulphate aluminium cement for 28 days is 4.0 Mpa. The fineness modulus means the index of the radius of river sand, and the average particle size is about 0.4 mm. Tex is a linear density unit, it refers to the weight of 1000 meters long yarn at the nominal moisture regain grams.
Comment 2: Improver use of the past tense (very frequent in the manuscript):....was shown in Figure (should be ...is shown...)
Thank you for your suggestion, we have improved the use of the past tense according to your suggestion.
Comment 3: table 3. the caption is very incomplete
Thank you for pointing this out, we have updated the manuscript on lines 174-176. The caption of Table 3 has been added, NC represents ordinary Portland cement and SSC represents sulphate aluminium cement in the table.
Comment 4: Section 3.2 title: compress-> compression
Thank you for your careful observation and correction, we have made changes on line 193.
Comment 5: Figure 3: caption icnomplete. One needs to go to the text to understand what is shown. Type of cement, meaning of sample names
Thank you for pointing this out, we have modified figure 3 and added a description of the details of the specimen.
Comment 6: Figure 4: 108 m 113 m, should be mm?
We thank you for pointing out this mistake, we have modified figure 4. Due to the limitation of the size of the text box in the picture, the text display is incomplete, and this problem has been solved.
Comment 7: Eq. 3 needs more details as to its derivation
We thank you for your suggestion, we added an explanation for Eq 3 in lines 304-309. We assume that the lateral expansion will occur during the compression of concrete, resulting in tensile stress on the uncompressed side of the concrete. The height of the necking position is determined by the deformation energy on both sides of the necking. We assume that the energy balance generated by the deformation on both sides of the necking position can obtain the equation about the height of the necking position as shown in Eq 3.
Comment 8: Table 5: compression strength means compression stress to rupture? Please explain
Thank you for your question. The compressive strength refers to the maximum compressive stress value of the specimen during compression. After reaching the compressive strength, the concrete will have a certain residual strength, so it is found in the curve that there is a falling section in the curve after the highest point.
Comment 9: Lines 199-204 are confusing: ...little effect....compresion could be improved. Is there improvement or not?
Thank you for your question, we have made changes on lines 212 and 215. The type of textiles has no significant effect on the compressive strength of ordinary Portland cement matrix, but has a significant effect on the compressive strength of self-stressing cement matrix. We have improved the description.
Comment 10: Line 208: why 2 MPa?
Thank you for your question, we have made changes on lines 220-221. The stress in the σ1 and σ2 directions is the self-stress inside the matrix. The measured self-stress is between 1.1MPa and 2.5MPa, so σ1=σ2=2MPa is selected.
Comment 11: Lines 222-226 also confusing. please rephrase in amore accessible manner.
Thank you for pointing this out, we have made changes on lines 233-236. We have changed the description of the theoretical and actual values of triaxial compression and biaxial compression.
Comment 12: Figure 4: we see 4 pictures, but nothing is explained in the caption. Label a,b,c,d and explain, please
We thank you for your valuable suggestion, and we have updated figure 4. These 4 pictures show the different stages of the specimen during compression, such as before cracking, crack initiation, crack development, and crack limit stage.

Reviewer 3 Report
The work is average but may be improved by the inclusion of the following suggestions.
1. English should be improved throughout the manuscript.
2. Quantitative information should be provided in the abstract.
3. The concussion should be concise and to the point indicating the application of the work.
4. The novelty of the work should be established.
5. Please write one paragraph in the introduction onto advanced materials, in general, and you can consult the following articles to make this manuscript more useful to the readers.
Chem. Select 4, 12708-12718 (2019).; J. Mol. Liq., 279, 251-266 (2019); Int. J. Biol. Macromol., 132, 244-253 (2019).; Aircr. Eng. & Aerosp. Technol., 92, 1027-1035 (2020).;
6. Please insert bars on the graphs wherever are required.
7. Improve Figure quality.
Author Response
Comment 1: English should be improved throughout the manuscript.
Thank you for your suggestion, we have modified some expressions in the manuscript.
Comment 2: Quantitative information should be provided in the abstract.
Thank you for your suggestion, we have added quantitative analysis and experimental data on lines 15-16 and 19-20.
Comment 3: The concussion should be concise and to the point indicating the application of the work.
Thank you for pointing this out. We have made changes on lines 355-357. We add a description of the practical application of this study in the conclusion. This manuscript provides a new idea for the fabrication of TRSSC, and provides a reference for the fabrication of TRSSC stress components and reinforcement components.
Comment 4: The novelty of the work should be established.
Thank you for your suggestion, we have updated the manuscript on lines 353-357. The innovation and practical significance of this article have been explained. We explored the influence of the change of textile types on the compressive properties of the matrix, and provided a reference for the application of 3D TRSSC.
Comment 5: Please write one paragraph in the introduction onto advanced materials, in general, and you can consult the following articles to make this manuscript more useful to the readers.
Chem. Select 4, 12708-12718 (2019).; J. Mol. Liq., 279, 251-266 (2019); Int. J. Biol. Macromol., 132, 244-253 (2019).; Aircr. Eng. & Aerosp. Technol., 92, 1027-1035 (2020).;
We thank you for your suggestion, we have made changes on lines 35-39. We have added an introduction to advanced materials and cited papers recommended by reviewers.
Comment 6: Please insert bars on the graphs wherever are required.
Thank you for your question, we have changed figure 3, and the bar graph has been inserted.
Comment 7: Improve Figure quality.
Thank you for pointing this out, we have improved the figures, such as figure 3 and figure 5.

Round 2
Reviewer 1 Report
My concerns are fully addressed by the authors. I recommend the publication of this manuscript.
Reviewer 3 Report
Accepted